# The Clinical Implications of Inappropriate Therapy in Community-Onset Urinary Tract Infections and the Development of a Bayesian Hierarchical Weighted-Incidence Syndromic Combination Antibiogram

**DOI:** 10.3390/antibiotics14020187

**Published:** 2025-02-12

**Authors:** Adolfo Gómez-Quiroz, Brenda Berenice Avila-Cardenas, Judith Carolina De Arcos-Jiménez, Leonardo Perales-Guerrero, Pedro Martínez-Ayala, Jaime Briseno-Ramirez

**Affiliations:** 1Microbiology Laboratory, Hospital Civil de Guadalajara “Fray Antonio Alcalde”, Guadalajara 44280, Mexico; agomezq@hcg.gob.mx (A.G.-Q.); 2019122@mail.hcg.udg.mx (B.B.A.-C.); 2Laboratory of Microbiological, Molecular and Biochemical Diagnostics (LaDiMMB), Tlajomulco University Center, University of Guadalajara, Tlajomulco de Zuñiga 45641, Mexico; judith.dearcos@academicos.udg.mx; 3Department of Internal Medicine, Hospital Civil de Guadalajara “Fray Antonio Alcalde”, Guadalajara 44280, Mexico; leonardo.perales6284@alumnos.udg.mx; 4HIV Unit, Hospital Civil de Guadalajara “Fray Antonio Alcalde”, Guadalajara 44280, Mexico; pedro.martinez@cucs.udg.mx; 5Health Division, Tlajomulco University Center, University of Guadalajara, Tlajomulco de Zuñiga 45641, Mexico

**Keywords:** antibiotic resistance, urinary tract infections, antimicrobial coverage, WISCA, weighted-incidence syndromic combination antibiograms, antibiotic stewardship, Bayesian analysis

## Abstract

Background/Objectives: The rise in multidrug-resistant pathogens complicates UTI management, particularly in empirical therapy. This study aimed to develop and describe a Bayesian hierarchical weighted-incidence syndromic combination antibiogram (WISCA) model to optimize antibiotic selection for adult patients with community-onset UTIs. Methods: A retrospective study was conducted using a Bayesian hierarchical model. Data from microbiology laboratory records and medical databases were analyzed, focusing on age, prior antibiotic exposure, and clinical characteristics. Clinical outcomes, including extended hospital stays and in-hospital mortality, were evaluated before WISCA model development. Unlike traditional antibiograms, a WISCA integrates patient-specific factors to improve antimicrobial coverage estimations. A total of 11 monotherapies and 18 combination therapies were tested against 15 pathogens, with posterior coverage probabilities and 95% highest density intervals (HDIs) used to assess coverage. Results: Inappropriate final antibiotic treatment was associated with worse outcomes. The Bayesian framework improved estimations, particularly for rare pathogen–antibiotic interactions, increasing model applicability in high-resistance settings. Combination regimens showed superior coverage, especially in MDR cases and older adults. Conclusions: This study employed a comprehensive methodological approach for WISCA development, enhancing empirical antibiotic selection by incorporating local resistance data and patient-specific factors in a middle-income Latin American country with a high antimicrobial resistance profile. These findings provide a foundation for future clinical applications and antimicrobial stewardship strategies in high-resistance environments.

## 1. Introduction

Urinary tract infections (UTIs) are common bacterial infections, with an incidence of 13,852.9 [12,135.6–15,480.3] per 100,000 population in our region, and they can lead to significant morbidity if not treated properly [1,2]. In outpatient settings, the increasing resistance to commonly used oral antibiotics poses a challenge [2]. Patients who receive antibiotics to which the pathogen is resistant are more likely to require additional antibiotic courses or hospitalization [2,3]. Community-onset urinary tract infections (CoUTIs) that require hospitalization often present with a challenging resistance profile and significant clinical consequences [3]. *Escherichia coli* is the most common uropathogen in CoUTIs, followed by *Klebsiella pneumoniae*, *Enterococcus* spp., and *Pseudomonas aeruginosa* [4]. Resistance to commonly used antibiotics such as ampicillin, cefazolin, cefuroxime, and co-trimoxazole is frequently observed, complicating the choice of empirical therapy [5]. In particular, extended-spectrum cephalosporin-resistant Enterobacterales (ESC-R EB) are associated with poor clinical outcomes, including increased rates of clinical failure and inappropriate initial antibiotic therapy (IIAT) [3].

The global burden of antimicrobial resistance (AMR) in UTIs is exacerbated by the widespread use of antibiotics, which accelerates the development of resistance mechanisms, such as β-lactamase production [6,7,8,9]. This issue is particularly pronounced in community-acquired UTIs, where resistance to fluoroquinolones has been documented to increase significantly over time in various regions, including Europe, Asia, and North America [10]. Similarly, patients with CoUTIs caused by multidrug-resistant (MDR) bacteria often experience IIAT, resulting in prolonged hospital stays and increased healthcare costs [11]. A significant proportion of uropathogens are MDR, with rates exceeding 40% in some studies [12]. ESBL-producing strains are increasingly common, particularly among *E. coli* and *K. pneumoniae* [13]. The presence of carbapenem-resistant (CR) organisms further exacerbates these issues, as these infections are associated with longer hospital stays, higher readmission rates, and increased mortality, especially in cases with concurrent bacteremia [14].

The SENTRY Antimicrobial Surveillance Program and other studies have documented these resistance patterns, emphasizing the need for continuous surveillance and tailored antimicrobial stewardship programs to manage and mitigate the impact of AMR in UTIs across Latin America [15,16]. Additionally, the presence of quinolone-resistant and ESC-R EB in both community and hospital settings, with resistance rates reaching up to 64% and 49%, respectively, further emphasizes the significant public health challenge posed by AMR in our country [16]. This heterogeneity in local epidemiology and bacterial resistance patterns complicates the selection of appropriate empirical antibiotics, reinforcing the need for customized tools based on local data [9,12,13,17].

The development of weighted-incidence syndromic combination antibiograms (WISCAs) has led to significant advancements in empirical antibiotic treatment, especially in the context of multidrug-resistant organisms (MDROs) [17,18]. Traditional antibiograms, while useful, often lack syndrome specificity and fail to consider the variability in bacterial prevalence across different populations and age groups [17]. Unlike traditional antibiograms, WISCAs provide tailored recommendations by weighting the coverage probability according to pathogen prevalence [17]. This methodology has shown potential in various clinical scenarios, including pediatric urinary tract infections [19], healthcare-associated urinary tract infections [20,21], neonatal sepsis [22], neonatal meningitis [23], febrile neutropenia in oncologic pediatric patients [24], ventilator-associated pneumonia [18], and pediatric bloodstream infections [25].

WISCAs incorporating Bayesian hierarchical models, which adjust for patient-specific factors such as age, sex, or prior antibiotic exposure, provide further refinement of these coverage estimates [21,24,26]. This methodology, adaptable to various clinical scenarios involving infectious syndromes, offers probability distributions for coverage adjusted to local resistance patterns and accounts for patient-specific factors, resulting in more accurate and tailored antibiotic recommendations [19,24,26]. Integrating WISCAs into clinical practice supports antimicrobial stewardship by providing timely, evidence-based guidance for antibiotic regimen selection, potentially reducing resistance and preserving antibiotic efficacy [19,22,24].

Most of the previous studies on WISCA design have been conducted in high-income countries with varying levels of AMR [17,18,19,20,21,22,23,24,25,26,27]. This study aims to address the knowledge gap regarding WISCA development for UTIs in middle-income Latin American countries, where high rates of AMR among Enterobacterales are prevalent [15,16]. Specifically, our objective was to evaluate the impact of inappropriate antimicrobial therapy in patients with community-onset UTIs who required hospitalization at our institution and to develop a WISCA designed to enhance the appropriate use of antibiotics, thereby supporting antimicrobial stewardship initiatives.

## 2. Results

According to the specified criteria, 242 episodes of patients with CoUTIs who presented to the emergency department were documented, as depicted in Figure 1.

The median age of the patients was 53.0 years (IQR 40.0–63.0), and 68.18% were female (*n* = 165). Among the documented CoUTI events, 99.17% were complicated urinary tract infections (*n* = 240), and 97.11% were reported as pyelonephritis (*n* = 235). Permanent urinary catheters were used in 11.57% of patients due to underlying anatomical or functional conditions (*n* = 28). The median Charlson comorbidity index was 3 (IQR 1–5), and the median SOFA score was 2 (IQR 0–6). Hypotension associated with infection was observed in 23.14% of patients (*n* = 56), and 18.59% required vasopressors (*n* = 45). The median length of hospital stay was 6.5 days (IQR 4–10), with the 90th percentile of hospital stays at 15 days. The in-hospital mortality rate was 14.05% (*n* = 34). The remaining sociodemographic data, comorbidities, and clinical characteristics are presented in Table 1.

In terms of local epidemiology and antibiotic resistance, the most frequently isolated organisms were *Escherichia coli* (71.07%, *n* = 172), *Klebsiella pneumoniae* (9.91%, *n* = 24), *Pseudomonas aeruginosa* (4.54%, *n* = 11), and *Proteus mirabilis* (2.89%, *n* = 7). According to standard definitions for acquired resistance, 46.44% of Enterobacterales isolates were classified as MDR (*n* = 98), and 61.61% of the isolates were resistant to third-generation cephalosporins (*n* = 130). No carbapenem-resistant Enterobacterales were identified. The rest of the isolated organisms and their resistance profiles are detailed in Table 2.

The most frequently used initial empirical therapy regimens were ertapenem (35.95%, *n* = 87), ceftriaxone (24.38%, *n* = 59), meropenem (9.92%, *n* = 24), and levofloxacin (8.68%, *n* = 21). Other regimens were used in 21.07% (*n* = 51) of the patients. The remaining antimicrobial regimens are listed in Appendix A. The initial empirical therapy was inappropriate for 29.75% of the UTI episodes (*n* = 72). However, in 91.74% of the UTI events, patients eventually received an appropriate regimen sometime during the course of the UTI episode. The median number of days taken to correct the initial inappropriate empirical treatment was 3 days (IQR 2–4) (Table 1).

According to the univariate analyses of in-hospital mortality and extended hospital stay, an older age, higher SOFA scores, the presence of hypotension, the need for vasopressors, a higher Charlson comorbidity index, the development of acute kidney injury (AKI) during the UTI event, a history of cardiovascular disease, chronic kidney disease, a previous cancer diagnosis, and complications related to urinary tract infection were positively associated with in-hospital mortality (*p* < 0.05). Additionally, the development of AKI during the UTI event and higher SOFA scores were positively associated with an extended hospital stay (*p* < 0.01). The remaining comparisons are detailed in Appendix A.

According to the multivariate logistic regression analyses regarding in-hospital mortality, two models were developed. Model A, which included inappropriate initial antibiotic treatment, identified higher SOFA scores (OR = 1.735, 95% CI: 1.348–2.396, *p* < 0.01) and the presence of complications related to urinary tract infection (OR = 7.207, 95% CI: 1.674–41.478, *p* < 0.05) as significant predictors of increased mortality, while the initial inappropriate antibiotic treatment was not significantly associated (OR = 1.439, 95% CI: 0.416–5.057, *p* = 0.563). Model B, which incorporated inappropriate final antibiotic treatment, revealed that inappropriate final antibiotic treatment was significantly associated with higher mortality (OR = 10.506, 95% CI: 1.311–135.265, *p* < 0.05), alongside elevated SOFA scores (OR = 1.719, 95% CI: 1.333–2.368, *p* < 0.01) and complications related to urinary tract infection (OR = 10.758, 95% CI: 2.218–84.270, *p* < 0.01). Both models demonstrated strong explanatory power (Pseudo R^2^ = 0.708 for Model A and 0.728 for Model B) and exhibited good fit (Hosmer–Lemeshow tests, *p* > 0.1 for both models). Figure 2a illustrates both models, while the complete set of variables for each model is provided in Appendix A.

Regarding factors associated with extended hospital stays, models were again developed using inappropriate initial empirical antibiotic treatment and inappropriate final antibiotic treatment as covariates. Model A, which included inappropriate initial antibiotic treatment, revealed that only AKI was significantly associated with extended hospitalization (Odds Ratio [OR] = 3.887; 95% Confidence Interval [CI]: 1.445–11.455; *p* < 0.01). Neither SOFA score (OR = 1.017; 95% CI: 0.891–1.155; *p* > 0.05), nor increasing age (OR = 1.012; 95% CI: 0.984–1.040; *p* > 0.05), nor initial empirical inappropriate treatment showed significant associations with extended stays. Model B, which incorporated inappropriate final antibiotic treatment, similarly identified AKI as a significant predictor of extended hospitalization (OR = 4.387; 95% CI: 1.640–13.013; *p* < 0.01). However, the SOFA score (OR = 1.013; 95% CI: 0.886–1.151; *p* > 0.05), age (OR = 1.010; 95% CI: 0.982–1.039; *p* > 0.05), and final inadequate treatment were not significantly associated with extended stays. Both models demonstrated modest explanatory power, with Pseudo R^2^ values of 0.129 for Model A and 0.135 for Model B, while exhibiting good fit according to the Hosmer–Lemeshow test (*p* > 0.1 for both models). Figure 2b illustrates both models, and the complete set of variables is provided in Appendix A.

Figure 3 shows the WISCA for hospitalized urinary tract infection (UTI) cases, comparing results obtained using a hierarchical Bayesian model and a non-hierarchical approach. For low-coverage agents such as ciprofloxacin (CIP), ceftazidime (CAZ), and ceftriaxone with vancomycin (CRO + VA), the Bayesian model produced even lower median estimates. This downward adjustment likely reflects the Bayesian model’s conservative handling of sparse data, which may “shrink” estimates toward the overall mean when evidence is limited. Conversely, for regimens with high static coverage in the traditional approach (red dots in Figure 3), such as carbapenem combinations (MEM + AN; ETP + AN), the hierarchical Bayesian estimates are slightly higher, with substantially narrower HDI ranges. For example, MEM + AN achieved a median coverage of 95.1% (HDI: 84.2–98.7%) under the Bayesian model, compared to the traditional static estimate of 93.4%. Similarly, ETP + AN’s Bayesian estimate was 95.1% (HDI: 84.2–98.6%), which aligns closely with traditional results but benefits from increased precision. Intermediate-coverage regimens such as nitrofurantoin (NIT) and aminoglycosides (ANs) demonstrated a similar pattern, with Bayesian estimates refining their static counterparts by providing narrower intervals. For instance, NIT’s coverage under the Bayesian model was 52.8% (HDI: 26.9–77.3%), reflecting greater uncertainty compared to AN, which showed a higher median coverage of 64.5% (HDI: 37.7–84.3%). The Bayesian WISCA approach also highlighted meaningful differences in regimens with overlapping ranges of effectiveness. For regimens like TZP + VA and TZP + LNZ, Bayesian estimates showed narrower HDIs (e.g., TZP + LNZ: median of 82.7%; HDI of 58.2–94.4%) compared to their static counterparts, emphasizing the hierarchical model’s advantage in accounting for variability across pathogen–regimen combinations. The complete set of medians of the posterior distribution and the associated 95% HDIs are shown in Figure 3 and in Appendix A.

The algorithm used to sample from the posterior distributions of the parameters achieved optimal convergence, with R^ index values near 1 and good mixing for all chains, as shown in Appendix A. Monte Carlo Markov chain (MCMC) trace plots of the WISCA model parameters and density plots of the posterior distributions for the general UTI population are depicted in Appendix A. Autocorrelation plots demonstrate minimal autocorrelation at distal lags (h ≥ 2), as shown in Appendix A.

The age-stratified WISCA is presented in Figure 4. Some clear differences were evident among the three age groups regarding the estimated coverage rates of various antibiotic regimens. Younger patients (<30 years) consistently demonstrated the highest coverage rates across most regimens, with carbapenem-based schemes (MEM, ETP) and their combinations (e.g., MEM + AN, ETP + AN) achieving coverage rates exceeding 90%, as reflected in the posterior median estimates. In contrast, the intermediate group (30–65 years) showed moderately lower coverage rates, with these same combinations maintaining favorable coverage (around 70–80%) but reflecting a decline in efficacy compared to younger patients. For patients over 65 years, the effectiveness of many regimens further declined, with some antibiotics—such as ceftriaxone (CRO)—showing notably low coverage levels, dropping below 40%. Moreover, specific regimens such as TZP + LNZ and TZP + VA also exhibited gradual reductions in efficacy from younger to older groups. Interestingly, fosfomycin (FOS) remained a relatively effective option even in older populations, with coverage rates nearing 70% in the >65-year group. The complete coverage data for regimens by age group are provided in Appendix A.

According to the univariate analyses of non-susceptibility to fluoroquinolones and third-generation cephalosporins in all organisms (Appendix A, respectively), factors related to comorbidities, disease severity, complications associated with urinary tract infections, and the presence of recurrent UTIs were significantly associated with the development of non-susceptibility in the general UTI population (*p* < 0.05). Building on these findings and considering the biological plausibility of factors associated with resistance, a WISCA was developed and stratified by subgroups to identify populations that could benefit from narrower-spectrum antimicrobials, while accounting for the resistance patterns observed in our population. The defined subgroups were based on the presence (and absence) of recurrent UTIs, antibiotic use within the previous 90 days, prior hospitalization (within 90 days), and UTI-related hypotension.

The analysis of WISCA across subgroups revealed important trends in antibiotic regimen coverage, despite the 95% HDIs overlapping in comparisons. For regimens with lower median coverage, non-recurrent urinary tract infections (UTIs) consistently showed higher median coverage compared to recurrent cases. For instance, ciprofloxacin (CIP) had a median coverage of 32.9% (HDI: 21.6–44.5%) in non-recurrent UTIs versus 29.4% (HDI: 18.0–41.2%) in recurrent cases. Similarly, piperacillin/tazobactam with vancomycin (TZP + VA) exhibited higher median coverage in non-recurrent cases (median: 43.5%; HDI: 33.3–52.8%) compared to recurrent ones (median: 35.9%; HDI: 24.5–46.9%). For high-coverage regimens, such as carbapenem-based combinations (e.g., MEM + AN), recurrent UTIs often showed slightly higher median coverage compared to non-recurrent cases, as in the case of MEM + AN (80.6% vs. 75.1%, respectively). While the overlapping HDIs indicate that these differences may not be statistically significant, the trends observed highlight potential variability influenced by regimen–pathogen interactions.

Subgroup analyses also highlighted the influence of prior hospitalization and recent antibiotic exposure on regimen coverage. The analysis of WISCA based on prior hospitalization status revealed a trend where antibiotic regimens generally demonstrated higher median coverage in patients without a history of hospitalization compared to those with prior hospitalization. For example, fosfomycin (FOS) showed a median coverage of 72.5% (HDI: 67.7–78.9%) in patients without prior hospitalization, compared to 65.5% (HDI: 55.8–75.8%) in those previously hospitalized. Similarly, nitrofurantoin (NIT) and aminoglycosides (ANs) displayed better coverage in patients without a hospitalization history, with median coverage rates of 60.1% (HDI: 52.9–66.9%) and 63.6% (HDI: 57.4–68.9%), respectively, compared to 56.3% (HDI: 46.8–65.5%) and 61.8% (HDI: 53.5–70.4%) in those with prior hospitalization. However, the 95% HDIs overlapped in most cases, indicating that these differences may not be statistically significant and should be interpreted cautiously. Similarly, prior antibiotic treatment within 90 days was associated with reduced coverage for multiple regimens. For instance, the coverage of piperacillin/tazobactam (TZP) decreased from 69.09% (HDI: 64.21–75.34%) in patients without recent antibiotic exposure to 64.66% (HDI: 54.99–73.73%) in those recently treated.

The analysis of WISCA in patients presenting with hypotension due to urinary tract infection (UTI), with or without the need for vasopressors, revealed a consistent trend of lower median coverage across nearly all antibiotic regimens compared to those without hypotension. For example, fosfomycin (FOS) demonstrated a median coverage of 74.5% (HDI: 69.5–80.3%) in patients without hypotension, compared to 66.9% (HDI: 55.2–78.5%) in those with hypotension. Similarly, nitrofurantoin (NIT) showed a median coverage of 64.4% (HDI: 57.6–70.4%) in patients without hypotension, while it decreased to 53.3% (HDI: 42.0–64.3%) in those with this condition. Among regimens with high coverage, meropenem combined with aminoglycoside (MEM + AN) achieved a median coverage of 78.4% (HDI: 73.7–84.9%) in patients without hypotension, compared to 68.5% (HDI: 59.3–78.2%) in those with hypotension. Similarly, meropenem combined with linezolid (MEM + LNZ) demonstrated a median coverage of 78.4% (HDI: 73.7–84.9%) in patients without hypotension, dropping to 66.8% (HDI: 57.5–76.3%) in patients with hypotension. Ceftriaxone combined with aminoglycoside (CRO + AN) showed a similar pattern, with a median coverage of 67.9% (HDI: 62.8–72.7%) in patients without hypotension, compared to 55.8% (HDI: 46.3–65.3%) in those with this condition. While the 95% HDIs for most regimens overlapped between groups, the consistent trend of reduced coverage in patients with hypotension underscores the need for careful regimen selection in this clinical scenario. The WISCA across subgroups related to recurrent UTI, prior hospitalization, recent antibiotic use within 90 days, and hemodynamic stability is depicted in Figure 5 and detailed in Appendix A.

Finally, based on the previous results and to promote the rational use of antibiotics, two subgroups of patients were identified for the development of a specific WISCA. Group 1 consisted of patients without a history of recurrent UTI, prior hospitalization, or recent antibiotic use within 90 days, and who were hemodynamically stable. Group 2 included patients who were hemodynamically stable and did not have comorbidities. Overall, antibiotic coverage was higher in these subgroups, particularly for regimens that previously demonstrated lower coverage, such as piperacillin/tazobactam plus aminoglycoside (TZP + AN), ceftriaxone plus aminoglycoside (CRO + AN), and piperacillin/tazobactam (TZP). Of these subgroups, TZP + AN showed 73.85% coverage (HDI: 53.79–78.92%), CRO + AN reached 75.7% (HDI: 55.9–80.7%), and TZP exhibited improved coverage with a median of 79.7% (HDI: 59.3–85.7%). Despite the wide HDIs, these regimens demonstrated notable improvements in coverage compared to previous analyses. These findings suggest that these specific populations may benefit from regimens that were previously considered less effective, supporting the rational selection of antibiotics in targeted clinical scenarios. The full results are presented in Figure 6 and in Appendix A.

## 3. Discussion

In this study, the sociodemographic and clinical characteristics of adult patients hospitalized with urinary tract infections were described, along with the resistance profiles of the causative microorganisms, and the impact of inappropriate therapy was evaluated. A high frequency of antimicrobial resistance was observed, whether acquired or intrinsic to the microorganisms, particularly against fluoroquinolones (78.52%) and third-generation cephalosporins (62.40%), among the pathogens responsible for UTIs (Appendix A). Furthermore, our findings revealed that inappropriate final antibiotic treatment during a UTI episode, elevated SOFA scores, and UTI-related complications—both local and systemic—significantly increased the risk of in-hospital mortality, which is consistent with reports on UTI-related mortality [28,29]. To address this, a weighted-incidence syndromic combination antibiogram tool was developed, extending the framework of traditional hospital antibiograms by providing weighted coverage estimates based on the frequency of identified pathogens. This approach was implemented using a Bayesian hierarchical model with random effects for both pathogens and treatment regimens. Our study contributes to the limited understanding of WISCA design for UTIs in middle-income countries. To the best of our knowledge, this is the first WISCA designed specifically for community-onset UTIs in adult patients within a Latin American country. Previous WISCA efforts have primarily been conducted in high-income countries, particularly in Europe and the United States, or as part of international collaborations between research groups [17,18,20,24].

The resistance profile of our institution is positioned at the higher end of the spectrum commonly reported in the recent literature [1,6,15,16,30]. Specifically, the proportion of multidrug-resistant (MDR) Enterobacterales (46.44%) and the elevated resistance to third-generation cephalosporins (61.61%) are concerning from both clinical management and public health perspectives [31,32]. These findings align with the global trend of increasing antimicrobial resistance among Enterobacterales, which has been documented for fluoroquinolones, third-generation cephalosporins, and, more recently, carbapenems [31]. The high usage of carbapenems for initial empirical therapy (46.86%) may reflect both the perceived risk and actual likelihood of encountering resistant pathogens. However, such empirical strategies may inadvertently propagate further resistance if not closely monitored and guided by local microbiological surveillance [33]. The proportion of initial empirical regimens deemed inappropriate (29.75%) underscores a critical gap between prescribing practices and actual pathogen susceptibility patterns, reinforcing the need for timely culture and susceptibility testing to inform definitive therapy. These resistance rates emphasize the importance of antimicrobial stewardship programs, which should include active local surveillance, regular antibiogram updates, and targeted interventions (e.g., de-escalation strategies and strict infection control measures) [34].

However, in our multivariate regression analysis, initial inappropriate empirical treatment was not associated with increased mortality. Instead, other determinants, such as disease severity (SOFA score) and UTI-related complications, were associated with in-hospital mortality. This finding may be explained by the rapid correction of the antimicrobial regimen in most patients with initially inappropriate empirical treatment, with a median correction time of 3 days (IQR 2–4), the relatively young patient population (mean age of 53 years; IQR of 40–63), and fewer comorbidities (Charlson comorbidity index of 3; IQR of 1–5). In contrast, inadequate final antibiotic treatment during the UTI episode was associated with increased mortality, highlighting the importance of achieving an appropriate antimicrobial regimen over the course of the UTI episode [35]. These findings align with the international literature regarding disease severity and inadequate treatment as key risk factors for in-hospital mortality in UTIs [28,35,36,37]. Additionally, acute kidney injury was positively associated with an extended hospital stay in our multivariate model, a finding consistent with the international literature [38,39]. Neither initial inappropriate empirical treatment nor inadequate directed treatment during the UTI episode was associated with an extended hospital stay in our models.

From the WISCA design for the general UTI population, it can be inferred that the most commonly used therapies in our institution, for which automated sensitivity testing is routinely performed, have coverage rates ranging from 9.04% for ciprofloxacin to 95.13% for meropenem + amikacin. Due to the high incidence of third-generation cephalosporin-resistant Enterobacterales in our institution, fosfomycin- and carbapenem-based regimens demonstrated better coverage compared to non-carbapenem-based regimens.

The non-Bayesian WISCA design yielded coverage rates comparable to those of the Bayesian hierarchical WISCA model, with most coverages rates falling within the 95% HDI of the Bayesian model. However, certain discrepancies were observed in specific scenarios, where coverage rates fell outside the HDIs of their Bayesian counterparts. The Bayesian hierarchical WISCA approach generally produces different coverage estimates compared to the traditional (non-Bayesian) WISCA, as it explicitly accounts for inherent variability and correlations among samples or subpopulations through the “partial pooling” of information across strata. In contrast, the conventional WISCA approach treats observations independently, which may lead to the under- or overestimation of coverage for certain antibiotic regimens. For example, regimens such as piperacillin/tazobactam or fosfomycin exhibited markedly higher coverage under the Bayesian model, likely due to the hierarchical prior structure that moderates extreme estimates, shrinking them toward a more stable central tendency. Conversely, some regimens show lower coverage estimates in the Bayesian model when the conventional method overestimates their efficacy. These discrepancies highlight the Bayesian framework’s advantage of integrating prior knowledge with observed data, resulting in more robust interval estimates and better accounting for small sample sizes or heterogeneous resistance patterns across clinical isolates. This approach is particularly valuable for managing complex infections, such as UTIs requiring hospitalization, where local resistance patterns significantly influence pathogen susceptibility. As a result, Bayesian hierarchical modeling provides more adaptive and subgroup-specific coverage estimates, in contrast to the “static coverage” rates generated by traditional WISCA designs.

Regarding coverage across regimens in the general population, it is important to highlight the role of fosfomycin, which demonstrated high coverage and consistent performance, even in patients previously treated with antibiotics. Recent studies support fosfomycin as an effective option for treating urinary tract infections, including those caused by ESBL-producing bacteria, and as a step-down therapy for complicated cases [40,41]. Currently, fosfomycin is FDA-approved in the United States for the treatment of uncomplicated UTIs caused by *Escherichia coli* and *Enterococcus faecalis* in women, but it is not approved for complicated UTIs [42,43]. While fosfomycin is generally well tolerated and effective, its use in upper UTIs should be carefully considered, particularly in complex cases such as those involving kidney transplant recipients [44,45]. The potential emergence of resistance during treatment remains a concern, emphasizing the need for further research to optimize dosing regimens and confirm their efficacy in broader clinical settings [44]. Nevertheless, these findings position fosfomycin as a viable alternative in settings with a high prevalence of multidrug-resistant organisms [46,47].

Coverage estimates varied across age groups. Older adults (>65 years) exhibited reduced coverage for beta-lactam-based regimens, likely due to higher rates of comorbidities and resistant organisms, while younger patients (<30 years) generally showed higher coverage, possibly linked to fewer antibiotic exposures. This is consistent with the idea that antimicrobial resistance is indeed a significant concern in elderly populations, particularly in the context of UTIs. Several factors contribute to this increased resistance, including age, comorbidities, and prior antimicrobial use [48]. Combination therapies with carbapenems or piperacillin/tazobactam, particularly when paired with amikacin or linezolid, demonstrated strong coverage across all age groups.

Coverage also varied by study subgroup, with lower rates generally observed in patients with recurrent UTIs, recent hospitalization, or recent antibiotic use, likely reflecting the selection of resistant strains. Notably, patients with UTI-related hypotension consistently showed lower coverage compared to those without this condition, highlighting the judicious use of broad-spectrum regimens in such cases [49,50,51]. Broad-spectrum agents, such as carbapenems (with or without amikacin) and fosfomycin, consistently provided robust coverage across subgroups, underscoring the importance of considering prior antibiotic history and clinical status when selecting an empirical therapy [49].

Finally, based on previous analyses, two groups were defined to facilitate practical decision-making for clinicians initiating antimicrobial regimens in our institution, aiming to promote rational antibiotic use informed by microbiological evidence. Group 1 (patients without recurrent UTI, recent hospitalization, or recent antibiotic use, and who are hemodynamically stable) and Group 2 (patients without comorbidities who are also hemodynamically stable) consistently achieved higher coverage across most regimens compared to previously analyzed subgroups. This pattern likely reflects a lower baseline risk of encountering resistant organisms due to fewer predisposing factors. These findings could encourage clinicians to initially use regimens such as TZP, TZP + AN, CRO + AN, and FEP + AN, thereby sparing carbapenems, and later de-escalate to fosfomycin, particularly in clinically stable patients with a lower risk of MDR organisms causing UTIs.

Regarding the methodology, our WISCA design incorporated methodological enhancements in prior selection, hierarchical random effects, and model validation. Our approach employed structured hierarchical priors to stabilize estimates, hierarchical shrinkage priors to reduce variability in low-frequency pathogen–regimen combinations, and a more comprehensive validation strategy using optimized Hamiltonian Monte Carlo (HMC) sampling and posterior predictive checks. Additionally, our model accounted for patient-specific factors such as prior antibiotic exposure and hospitalization history, enhancing its adaptability to complex clinical settings. These refinements contribute to a more robust and interpretable WISCA model, optimizing empirical antibiotic selection in a region with high antimicrobial resistance rates.

The limitations of our study are primarily due to its retrospective nature. First, during data collection, some variables were missing and had to be excluded from the analyses. While strict inclusion criteria were applied to ensure data quality, retrospective designs inherently limit the ability to control for all potential confounders. Prospective validation studies are needed to further evaluate the robustness of our findings and their applicability in clinical decision-making.

Additionally, the single-center nature of this study may limit the generalizability of the results to other settings, particularly those with differing patient demographics, healthcare resources, or antimicrobial resistance patterns. However, this limitation aligns with one of the primary objectives of WISCAs: to generate an antibiogram tailored to specific local conditions, particularly in regions with high antimicrobial resistance. Future multi-center studies and external validation will be crucial to assess the model’s adaptability across diverse clinical settings.

Another limitation is that while the study relied on antimicrobial susceptibility testing, these results do not always directly translate to clinical efficacy, as they do not account for pharmacokinetic and pharmacodynamic factors that influence antibiotic effectiveness in vivo. Additionally, some pathogens lacked well-defined susceptibility breakpoints, requiring the use of documented extrapolations and assumptions, which are detailed in the Appendix A for transparency.

Finally, although the WISCA model is specifically designed to capture the dynamic interactions between antimicrobial regimens and pathogens, adjusting for patient-specific characteristics and clinical scenarios, its performance remains dependent on the quality and comprehensiveness of the input data. While the model accounts for key variables such as age, sex, and prior antibiotic exposure, it may not fully capture unmeasured or unknown confounding factors, such as underlying comorbidities, immune status, or variations in healthcare practices. However, one of the key advantages of a WISCA is its ability to generate meaningful coverage estimates even in settings with moderate sample sizes, where traditional antibiograms may lack clinical specificity.

The Bayesian hierarchical modeling approach applied in this study enables partial pooling across regimens and pathogens, stabilizing estimates even for low-frequency antibiotic–pathogen interactions. The iterative nature of Bayesian inference allows for the progressive refinement of coverage probabilities over thousands of iterations, ensuring that uncertainty is appropriately accounted for. Future studies should evaluate the real-world impact of WISCAs on empirical antibiotic selection, antimicrobial stewardship programs, and long-term resistance trends.

Overall, these findings underscore the significant challenges in managing antibiotic resistance. As a reference center serving the uninsured population in western Mexico, our university hospital unit encounters a higher incidence of patients requiring hospitalization for urinary tract infections compared to what is typically reported. A notable strength of this study is the development of WISCA models utilizing a Bayesian hierarchical framework, which were specifically tailored to the local epidemiology characterized by a high resistance profile. The analysis of posterior coverages and 95% highest density intervals (HDIs) revealed significant variability in antibiotic coverage, predominantly influenced by resistance patterns. These results emphasize the critical importance of carefully selecting empirical therapies and continuously monitoring and adapting antibiotic policies to combat resistance effectively. Moreover, the expansion of WISCA models to other infectious syndromes will further enhance the capacity for evidence-based empirical therapy regimens, addressing the heterogeneity of local epidemiology. These efforts collectively support antimicrobial stewardship by providing actionable tools to implement rational empirical antibiotic treatments, ultimately improving patient outcomes in our population.

## 4. Materials and Methods

### 4.1. Population and Eligibility Criteria

A retrospective review of medical records was conducted for patients admitted to the emergency department and medical wards of our university hospital with International Classification of Diseases, 10th Revision (ICD-10) codes N39.0 (Urinary Tract Infection), N30.0 (Acute Cystitis), and N10 (Acute Pyelonephritis). Urinary culture results from the hospital’s microbiology laboratory were analyzed, including samples obtained via clean-catch midstream or catheterization from patients presenting to the emergency department or hospitalized in medical wards. Only initial cultures reported as positive were included. Records from patients admitted between June 2021 and December 2024 were reviewed. Patients included in the analysis met the following criteria: they presented to the emergency department with community-onset symptoms of a UTI, their hospitalization was due to a UTI diagnosis, and they had a positive urinary culture at admission, which was defined as a UTI event. Patients younger than 18 years of age, those with more than 10% missing data from clinical records, and those with episodes of asymptomatic bacteriuria were excluded.

Demographic data, comorbidities, the Charlson comorbidity index (CCI), the Sequential Organ Failure Assessment (SOFA) score, and other clinical and microbiological parameters were systematically collected from patient medical records and microbiology registries. The collected data included variables such as the type of urinary infection (complicated versus uncomplicated), site of UTI (pyelonephritis or cystitis), use of a permanent urinary catheter, and presence of recurrent UTI (defined as at least three episodes within 12 months or at least two episodes within 6 months). UTI-related local complications (e.g., renal or perinephric abscess, emphysematous cystitis or pyelonephritis, xanthogranulomatous pyelonephritis, malakoplakia, and renal papillary necrosis) and systemic complications (e.g., bacteremia, UTI-related hypotension, and UTI-related need for vasopressors) were also recorded. Additional variables included the initial empirical treatment used, time to the correction of inappropriate initial antibiotic therapy, prior antibiotic use within 90 days, prior hospitalization within 90 days, presence of hypotension, need for vasopressors, in-hospital mortality, length of hospital stay, and susceptibilities of the isolated microorganisms. If the isolated microorganism was not susceptible to the initial empirical treatment, the event was classified as inappropriate initial antibiotic treatment. Furthermore, if inappropriate initial antibiotic treatment was not corrected based on urinary culture results during the UTI episode, it was classified as inappropriate final antibiotic treatment (IFAT).

For the type of UTI, complicated UTIs were defined based on the presence of functional, metabolic, or anatomical factors such as diabetes mellitus, immunocompromised status, nephrolithiasis, urinary obstruction, a neurogenic bladder, an indwelling urinary catheter or other device, and male sex [52,53,54].

The bacterial isolates were identified using standardized techniques and methods, and their antibiotic susceptibility was tested with the VITEK 2 system by Biomerieux (Marcy l’Etoile, France). Specific panels were used in accordance with Clinical and Laboratory Standards Institute (CLSI) guidelines [55]. Results were interpreted based on CLSI breakpoints, with extrapolations made for agents lacking established breakpoints or minimum inhibitory concentration (MIC) values, as detailed in Appendix A [55].

### 4.2. WISCA Development

The WISCA model was designed based on pathogens isolated from patients with UTIs. As previously described, a UTI event was considered if the patient presented symptoms consistent with a clinical syndrome of UTI and had a positive urine culture upon admission. This process was independently conducted by two infectious disease specialists, and any discrepancies were discussed and resolved through consensus. If the isolated organism was susceptible to an empirical antibiotic, it was reported as susceptible to the empirical antimicrobial regimen employed. Organisms with intermediate susceptibility to a particular antimicrobial were classified as resistant for the model design. In cases where more than one microorganism was identified in the same urine culture, it was considered contaminated and was subsequently excluded from the model [55]. For patients with multiple subsequent urine cultures, it was verified that these patients experienced a new UTI event. Subsequent cultures taken during the initial UTI episode were excluded to avoid results influenced by pharmacological pressure. If they corresponded to a new UTI event, they were included in the analysis.

The coverage of regimens was studied according to local and international guidelines, including regimens for which automated sensitivity testing is routinely performed [55,56,57]. These regimens included fosfomycin (FOS), nitrofurantoin (NIT), trimethoprim/sulfamethoxazole (SXT), amikacin (AN), ciprofloxacin (CIP), piperacillin/tazobactam (TZP), ceftazidime (CAZ), ceftriaxone (CRO), cefepime (FEP), ertapenem (ETP), and meropenem (MEM). Additionally, combination therapies, including ceftazidime + amikacin (CAZ + AN), ceftriaxone + amikacin (CRO + AN), piperacillin/tazobactam + amikacin (TZP + AN), cefepime + amikacin (FEP + AN), meropenem + amikacin (MEM + AN), ertapenem + amikacin (ETP + AN), ceftazidime + linezolid (CAZ + LNZ) or vancomycin (CAZ + VA), ceftriaxone + linezolid (CRO + LNZ) or vancomycin (CRO + VA), piperacillin/tazobactam + linezolid (TZP + LNZ) or vancomycin (TZP + VA), cefepime + linezolid (FEP + LNZ) or vancomycin (FEP + VA), ertapenem + linezolid (ETP + LNZ) or vancomycin (ETP + VA), and meropenem + linezolid (MEM + LNZ) or vancomycin (MEM + VA), were studied.

### 4.3. Statistical Analysis

Demographic data are reported as simple relative frequencies. The normality of the data distribution was assessed via the Shapiro–Wilk test. Pearson’s chi-square test and Fisher’s exact test were used to compare proportions as appropriate. For comparisons of quantitative variables, Student’s t tests and Wilcoxon–Mann–Whitney tests were used for normally and non-normally distributed data, respectively.

To identify the impact of an inadequate antibiotic regimen on clinical outcomes, a multivariate logistic regression model was used. The clinical outcomes were in-hospital mortality and extended hospital stays, defined as a stay exceeding the 90th percentile of our patient population. A stepwise method was used for variable selection, ensuring the inclusion of relevant predictors while optimizing model performance. The Hosmer–Lemeshow test was applied to determine the goodness of fit, with values greater than 0.1 considered appropriate.

Univariate analyses were performed on demographic, clinical, and comorbidity factors with biological plausibility for the presence of resistance to identify factors associated with antimicrobial resistance and determine subgroups where the WISCA tool could be applied to promote the rational use of antibiotics. Fluoroquinolone non-susceptibility and third-generation cephalosporin non-susceptibility were used as dependent variables in the entire set of patients (and, consequently, all microorganisms), given their established role as markers of resistance in microorganisms associated with UTIs, in the development of a WISCA (which considers all organisms causing a particular infectious syndrome), based on the methodologies described [16,17,26,58,59]. Variables identified as associated with resistance, along with those with biological plausibility, were subsequently used in the exploration of subgroups within the WISCA framework.

Finally, in accordance with previous studies [17,18,20,24,26], a Bayesian hierarchical logistic regression model was employed to estimate the probability that a given antibiotic regimen covers a specific pathogen causing a urinary tract infection (UTI). Covariates for age categories (<30 years, 30–65 years, and >65 years) and sex were included to account for potential variability. In particular, the following were introduced:Fixed effects for age strata (<30 years, 30–65 years, and >65 years) and sex, capturing systematic coverage differences across these patient subgroups;Random effects for both regimen and pathogen, allowing each regimen and pathogen to share information with one another, thereby reducing the uncertainty in regimens or pathogens with few observations.

Such an approach extends the weighted-incidence syndromic combination antibiogram (WISCA) framework by combining the partial pooling advantage with a straightforward means to incorporate patient covariates.

Model Specification

Let *i* = 1, …, *N* index the observations, each corresponding to a patient with covariates, a chosen antibiotic regimen *j*[*i*], and an identified pathogen *k*[*i*]. Define the following:
coverage*_i_ ∈* {0,1}: an indicator that regimen *j* successfully covers pathogen *k* in the *i*-th observation.Age and sex: patient-level covariates (age group and sex).α_0_: overall intercept.α_pathogen[*k*]_: random intercept for pathogen *k*.α_regimen[*j*]_: random intercept for regimen *j*.*β*_1_,*β*_2_: fixed-effect coefficients for age group and sex, respectively.

The probability of coverage was modeled via a Bernoulli likelihood with a logit link:\logit(P(coverage*_i_* = 1)) = α_0_ + α_pathogen[*k*[*i*]]_ + α_regimen[*j*[*i*]]_ + *β*_1_ age group*_i_* + *β*_2_ sex*_i_*.

Hence,*P*(coverage*_i_* = 1) = logit^−1^ (α_0_+ α_pathogen[*k*[*i*]]_ + α_regimen[*j*[*i*]]_ + *β*_1_ age group*_i_* + *β*_2_ sex*_i_*).

Random effects were assumed as follows:α_pathogen[*k*]_ ~ *N* (0,α_pathogen_), α_regimen[*j*]_ ~ *N*(0,α_regimen_).

Prior Distributions

Moderately informative priors were placed on model parameters, ensuring stability while allowing the data to drive estimates:Intercept (α_0_): e.g., α_0_ ~ Student-*t*(*ν* = 3, *μ* = 0, *σ* = 10).Fixed effects (*β*_1_, *β*_2_): e.g., *β* ~ Student-*t*(3,0,1) or normal(0,1).Random effects’ standard deviations (α_pathogen_, α_regimen_): e.g., half-Student-*t* or half-normal, such asα_pathogen_ ~ normal^+^(0,0.3), α_regimen_ ~ normal^+^ (0,0.3).

These priors regularized extreme intercepts and kept the model stable when data were sparse in certain pathogen–regimen combinations.

Missing data were handled using a combination of complete-case analysis and Bayesian hierarchical modeling, which allows for partial pooling across regimens and pathogens to generate stable estimates despite sparse observations.

The Hamiltonian Monte Carlo (HMC) algorithm, implemented via Stan software (https://mc-stan.org/), was used for Bayesian inference. The algorithm executed four chains, each with 10,000 iterations, discarding the first 3000 iterations as a warm-up period. Convergence was assessed using trace plots, R-hat diagnostic indices, and autocorrelation evaluations.

The parameters and coverage estimates are presented as medians accompanied by 95% highest density intervals (HDIs), where broader HDIs reflect higher levels of uncertainty. These summaries are robust to potential skewness in the posterior distributions, providing reliable and informative estimates of antibiotic regimen coverage. Autocorrelation for the parameters was evaluated at distal lags (e.g., between Xt and Xt + h, where h ≥ 2) to ensure the independence of samples across iterations.

The statistical analysis was conducted using R software (version 4.3.0). Data management and manipulation were primarily performed with the *dplyr* and *tidyr* packages. The Bayesian hierarchical models were fitted via *brms* (version 2.19.0), which interfaces with the Stan framework for model specification and Hamiltonian Monte Carlo sampling. Model diagnostics and posterior summaries were completed using the *posterior*, *bayesplot*, and *tidybayes* libraries (versions 1.3.0, 1.10.0, and 3.0.5, respectively), ensuring the rigorous evaluation of convergence and robust inference.

## 5. Conclusions

This study highlighted the significant challenges posed by antimicrobial resistance in the treatment of urinary tract infections (UTIs) in a middle-income Latin American context. A Bayesian hierarchical weighted-incidence syndromic combination antibiogram model was developed, tailored to the resistance profiles of the study population, providing a structured methodological approach for optimizing empirical antibiotic selection. However, it is important to acknowledge that this study focused on the development of the WISCA model rather than its direct clinical implementation. While the model demonstrated the potential to generate subgroup-specific antimicrobial coverage estimates, its real-world effectiveness in improving clinical outcomes and guiding antimicrobial stewardship has yet to be validated. The findings revealed a high prevalence of multidrug-resistant pathogens, particularly Enterobacterales, and underscored the impact of inappropriate final antibiotic treatment on in-hospital mortality. These results emphasized the need to incorporate local epidemiological data into antibiotic policies and explore tools like WISCAs as potential evidence-based decision-making aids. Future research should focus on prospective validation, multi-center evaluations, and implementation studies to assess the WISCA’s impact on antibiotic prescribing, patient outcomes, and long-term resistance trends. Additionally, the periodic recalibration of the model will be necessary to ensure its adaptability to evolving resistance patterns. In conclusion, while the WISCA model represents a methodological advancement, its clinical utility should be evaluated through rigorous prospective studies before widespread implementation in empirical antibiotic therapy.

## Figures and Tables

**Figure 1 antibiotics-14-00187-f001:**
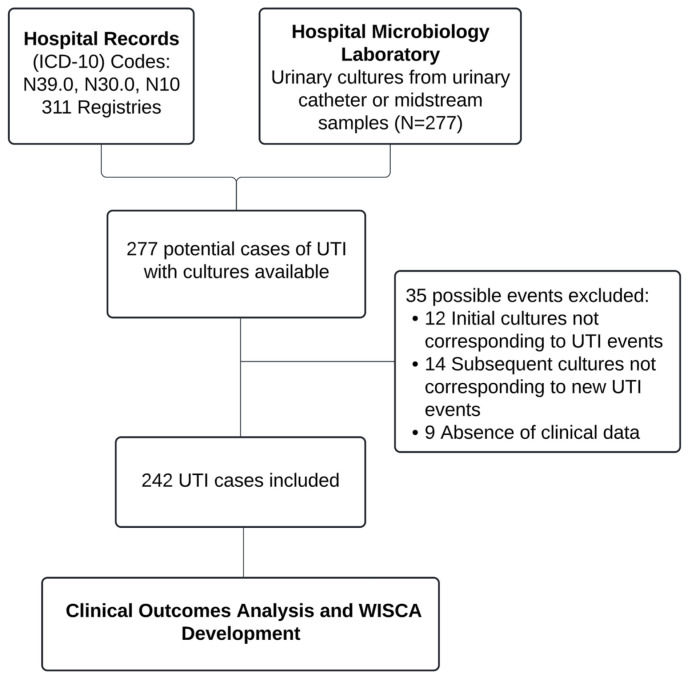
Flowchart of case selection.

**Figure 2 antibiotics-14-00187-f002:**
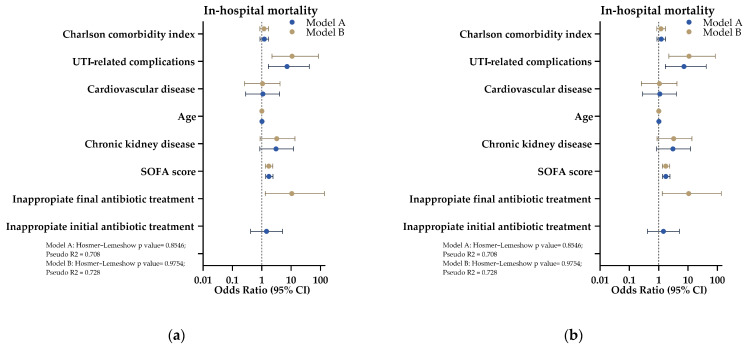
Logistic regression analysis for clinical outcomes. Model A includes initial empirical inappropriate therapy (blue dots and lines) as a covariate, while Model B includes inappropriate final treatment (red dots and lines) as a covariate. (**a**) Multivariate models for hospital mortality; (**b**) multivariate models for extended hospital stay.

**Figure 3 antibiotics-14-00187-f003:**
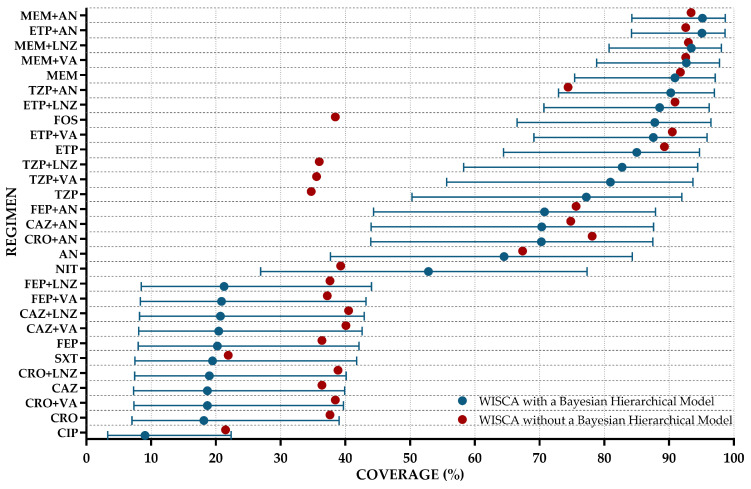
Weighted-incidence syndromic combination antibiogram for community-onset urinary tract infections. Blue dots represent the median of the posterior distribution of antibiotic regimens (coverage), and the blue lines indicate the 95% highest density interval (HDI). Red dots represent the WISCA without a Bayesian hierarchical model. Regimens are displayed in descending order of coverage.

**Figure 4 antibiotics-14-00187-f004:**
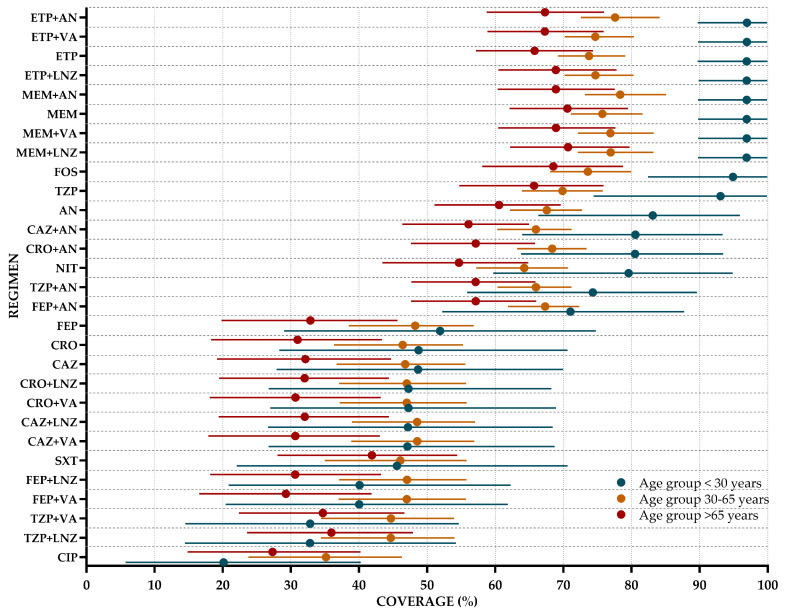
Weighted-incidence syndromic combination antibiogram for community-onset urinary tract infections across different age subgroups. Dots represent the median of the posterior distribution of antibiotic regimens (coverage), and the lines indicate the 95% highest density interval (HDI). Regimens are displayed in descending order of coverage for the subgroup under 30 years of age.

**Figure 5 antibiotics-14-00187-f005:**
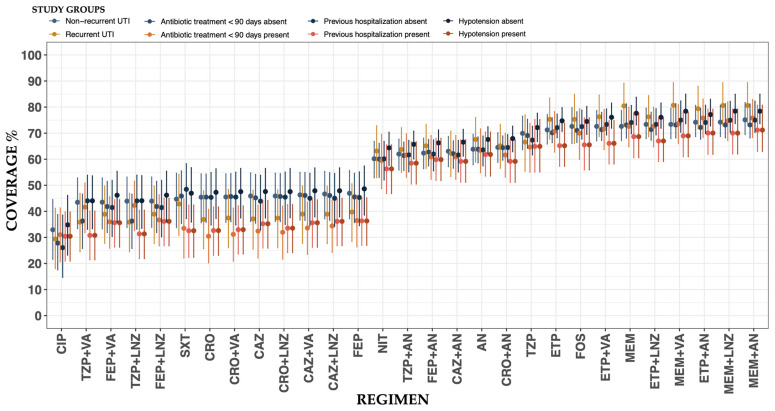
Weighted-incidence syndromic combination antibiogram for community-onset urinary tract infections across subgroups, ordered from left to right: recurrent UTI, prior hospitalization, recent antibiotic use within 90 days, and hemodynamic stability. The *X*-axis represents the treatment regimens, while the *Y*-axis indicates the percentage of posterior coverage (dots) and the 95% highest density intervals (HDIs) (lines).

**Figure 6 antibiotics-14-00187-f006:**
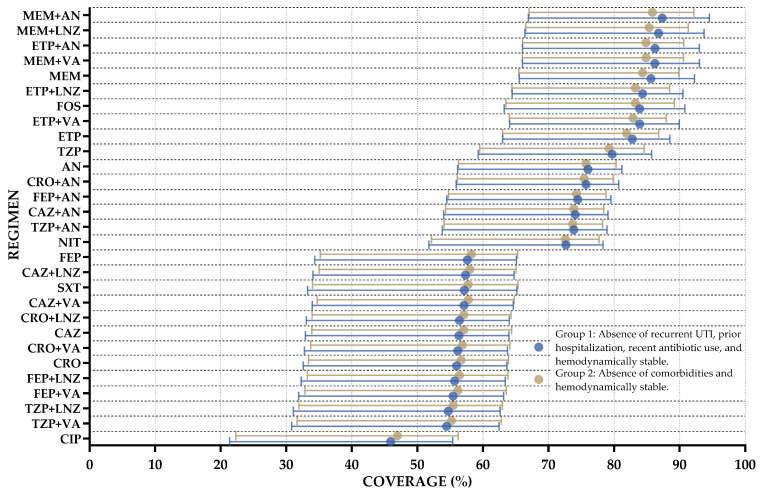
Weighted-incidence syndromic combination antibiogram (WISCA) for community-onset urinary tract infections across two subgroups with favorable resistance profiles. The *X*-axis represents the treatment regimens, while the *Y*-axis indicates the percentage of posterior coverage (dots) and the 95% highest density intervals (HDIs) (lines).

**Table 1 antibiotics-14-00187-t001:** Sociodemographic data, comorbidities, and clinical characteristics of patients who presented to the emergency department with community-onset UTIs.

Variable	Obs. (*n* = 242)	Total (*n* = 242)	Women (*n* = 165)	Men (*n* = 77)	Significance *
Sociodemographic and clinical characteristics					
Age—mean (IQR)	165/77	53 (40.0–63.0)	51 (38.0–63.0)	55.0 (47.0–67.0)	*
Complicated UTI—*n* (%)	165/77	240 (99.17)	163 (98.78)	77 (100.00)	NS
Permanent urinary catheter—*n* (%)	165/77	28 (11.57)	11 (6.66)	17 (22.08)	NS
Recurrent UTI—*n* (%)	165/77	93 (38.43)	58 (35.2)	35 (45.45)	NS
Pyelonephritis—*n* (%)	165/77	235 (97.11)	158 (95.8)	77 (100.00)	NS
Initial inappropriate antibiotic treatment—*n* (%)	164/75	72 (29.75)	44 (26.67)	28 (36.36)	NS
Days to correction of empirical treatment—median (IQR)	44/28	3.0 (2.0–4.0)	3.0 (2.0–4.0)	3.00 (2.75–4.0)	NS
Inappropriate final antibiotic treatment in UTI event—*n* (%)	164/77	20 (8.26)	11 (6.67)	9 (11.69)	NS
Previous antibiotic treatment within 90 days—*n* (%)	165/77	88 (36.36)	50 (30.30)	38 (49.35)	NS
Previous hospitalization within 90 days—*n* (%)	165/77	108 (44.63)	69 (41.82)	39 (50.65)	NS
Severity of illness					
Hypotension—*n* (%)	165/77	56 (23.14)	37 (22.42)	19 (24.68)	NS
Need for vasopressors—*n* (%)	165/77	45 (18.59)	31 (18.79)	14 (18.18)	NS
SOFA score—median (IQR)	165/77	2 (0.0–6.0)	2 (0.0–6.0)	3.0 (0.0–7.0)	NS
Comorbidities	165/77	190 (78.51)	129 (78.18)	61 (79.22)	NS
Charlson comorbidity index—median (IQR)	165/77	3 (1.0–5.0)	3 (1.0–5.0)	3 (2.0–5.0)	NS
Diabetes mellitus—*n* (%)	165/77	150 (61.98)	105 (63.64)	45 (58.44)	NS
Hypertension—*n* (%)	165/77	82 (33.88)	62 (37.58)	20 (25.97)	NS
Cardiovascular disease—*n* (%)	165/77	48 (19.83)	35 (21.21)	13 (16.88)	NS
Acute kidney injury—*n* (%)	165/77	105 (43.39)	65 (39.39)	40 (51.95)	NS
Chronic kidney disease—*n* (%)	165/77	68 (28.10)	49 (29.70)	19 (24.68)	NS
Chronic liver disease—*n* (%)	165/77	4 (1.65)	3 (1.82)	1 (1.30)	NS
Pregnancy—*n* (%)	165/77	9 (3.72)	9 (5.45)	0.0 (0.0)	NS
Immunosuppression—*n* (%)	165/77	14 (5.79)	10 (6.06)	4 (5.95)	NS
Cancer—*n* (%)	165/77	15 (6.20)	11 (6.67)	4 (5.95)	NS
Central nervous system neurological disease—*n* (%)	165/77	28 (11.57)	11 (6.67)	17 (22.08)	NS
Previous diagnosis of peripheral neuropathy—*n* (%)	165/77	26 (10.74)	23 (13.94)	3 (3.90)	*
Outcome					
Urinary tract infection-related complications	165/77	68 (28.10)	47 (28.48)	21 (27.27)	NS
Local complication of UTI—*n* (%)	165/77	29 (11.98)	20 (12.12)	9 (11.69)	NS
System complication of UTI—*n* (%)	165/77	48 (19.83)	34 (20.60)	14 (18.18)	NS
Hospital stay—median (IQR)	165/77	6.5 (4.0–10.0)	7.0 (4.0–10.0)	6.0 (3.0–12.0)	NS
Hospital stay > 15 days—*n* (%)	165/77	29 (11.98)	17 (10.30)	12 (15.56)	NS
In-hospital mortality—*n* (%)	165/77	34 (14.05)	24 (14.55)	10 (12.99)	NS

* Significance levels: *: *p* < 0.05; NS: not significant.

**Table 2 antibiotics-14-00187-t002:** Organisms isolated in patients with community-onset UTI during the study period.

Variable	Obs. (*n* = 242)	Total (*n* = 242)	Women (*n* = 165)	Men (*n* = 77)	Significance *
Microorganism—*n* (%)					
*Escherichia coli*	165/77	172 (71.07)	129 (78.18)	43 (55.84)	**
*Klebsiella pneumoniae*	165/77	24 (9.91)	14 (8.48)	10 (12.98)	NS
*Pseudomonas aeruginosa*	165/77	11 (4.54)	3 (1.81)	8 (10.38)	**
*Proteus mirabilis*	165/77	7 (2.89)	6 (3.63)	1 (1.29)	NS
*Candida glabrata*	165/77	6 (2.47)	5 (3.03)	1 (1.29)	NS
*Enterococcus faecalis*	165/77	5 (2.06)	1 (0.60)	4 (5.19)	*
*Citrobacter freundii*	165/77	4 (1.65)	0 (0.00)	4 (5.19)	*
*Staphylococcus aureus*	165/77	3 (1.23)	2 (1.21)	1 (1.29)	NS
*Candida tropicalis*	165/77	3 (1.23)	1 (0.60)	2 (2.59)	NS
*Morganella morganii*	165/77	2 (0.82)	0 (0.00)	2 (2.59)	NS
*Enterobacter cloacae*	165/77	1 (0.41)	1 (0.60)	0 (0.00)	NS
*Staphylococcus saprophyticus*	165/77	1 (0.41)	1 (0.60)	0 (0.00)	NS
*Acinetobacter baumannii complex*	165/77	1 (0.41)	1 (0.60)	0 (0.00)	NS
*Candida parapsilosis*	165/77	1 (0.41)	0 (0.00)	1 (1.29)	NS
*Enterococcus faecium*	165/77	1 (0.41)	1 (0.60)	0 (0.00)	NS
Resistance Profile—*n* (%) **					
MDR Enterobacterales	151/60	98 (46.44%)	70 (46.35)	28 (46.66)	NS
3rd Gen Cephalosporin-Resistant Enterobacterales	151/60	130 (61.61)	91 (60.26)	39 (65.0)	NS
MDR Pseudomonas	8/3	3 (27.27)	0 (0.00)	3 (100)	NS
MDR Enterococcus	2/4	1 (16.66)	1 (50.0)	0 (0.00)	NS
MRSA	2/1	2 (66.66)	2 (100)	0 (0.00)	NS

* Significance levels: **: *p* < 0.01; *: *p* < 0.05; NS: not significant. ** Proportions were calculated based on the total isolates of *Enterobacterales*, *Pseudomonas*, *Enterococcus*, and *Staphylococcus aureus*, respectively. MDR: multidrug-resistant; 3rd Gen Cephalosporin: third-generation cephalosporin; MRSA: methicillin-resistant *Staphylococcus aureus*.

## Data Availability

The raw data supporting the conclusions of this article will be made available by the authors on request.

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
