# Peer review of "The Clinical Implications of Inappropriate Therapy in Community-Onset Urinary Tract Infections and the Development of a Bayesian Hierarchical Weighted-Incidence Syndromic Combination Antibiogram"

_antibiotics, 2025, doi:10.3390/antibiotics14020187_

Round 1

Reviewer 1 Report

Comments and Suggestions for Authors

Dear Authors,

The document meets the necessary standards for publication, providing adequate and sufficient information. The schemes and figures effectively summarize the content. However, the work lacks novelty and does not significantly impact resistance treatment. The methods and tools presented are standard and widely used. While the document is solid and well-prepared, it represents an average contribution without offering innovative insights, and it is likely to generate limited interest among the journal's readership.

Author Response

We sincerely thank the reviewers for their time and effort in evaluating our manuscript. Their insightful suggestions and recommendations have significantly improved the quality of this work.

As part of this round of revisions, we have incorporated the requested corrections. Attached to this message, we provide a detailed description of the changes made, which have been highlighted in the revised manuscript for easy identification.

Reviewer 2 Report

Comments and Suggestions for Authors

It is an original study. The results are well presented and discussed. However, some minor revisions needed:

Line 24: Instead of “extracted” can be written “obtained” or “taken”.

Line 111, 144, 168, 195, 199, 205, 209, 233, 235, 244, 271, 318, 337, 416, 478, 499, 508, 552, 611, 663: Passive voice (sentences) should be used. Passive voice is used as the appropriate language in academic studies. Please take this into consideration.

Line 190: “p value of <0.1” ? Why was this needed, what is the reason?

Table 1, Table 2, Table S3, Table S4, Table S5, Table S8: The p value should be replaced by significant, and **, * and NS should be used. “**: p<0.01; *: p<0.05; NS: not significant”: This information should be given as a legend below the table. Also, p<0.001 is not needed in the tables. p<0.01 can be used instead of p<0.001. P<0.01 is a good statistical value.

Line 308: “extended2” ??

Line 319: Instead of “p <0.001” can be written “p<0,01”.

Line 321: Instead of “p = 0.013” must be written “p<0,05”.

Line 325: Instead of “p = 0.042” must be written “p<0,05”.

Line 326: Instead of “p <0.001” can be written “p<0,01”.

Line 327: Instead of “p = 0.008” must be written “p<0,01”.

Line 329: “p = 0.8546 and p = 0.9754,” There is no need to write these values. NS is sufficient. However, p>0.05 can also be written.

Line 341, 342, 343, 346, 347, 350, 351: P values in other sections should be adjusted as indicated above.

Author Response

(The authors gave the same response as above.)

Reviewer 3 Report

Comments and Suggestions for Authors

The authors should consider the followings:

The quality and quantity of data should be sufficient for the model development. The authors should clarify and justify how they assess the data quality. It is rather low in quantity for only 241 cases included. 

Please specify which dataset wete used as training dataset, and which dataset for the validation dataset .

The authors should justify and describe how they treat missing data in the model development.

The authors should specify the novelty of this article in the abstarct and/or in the conclusion part.

The authors should compare studies of this similar approaches from other studies, especially those comparable studies within the region or nation concerned.

Limitations of this study should be critically reviewed.

By the conclusion part, the authors should not overstate the power and usage of the model without well quality results and data for supporting.

Background hospital infection rate and regional infection rate should be provided. 

The authors should justify the measures for how they prevent overfitting of the data.

Comments on the Quality of English Language

Quality of English language should be enhanced 

Author Response

(The authors gave the same response as above.)
